# VISUAL IMITATION WITH REINFORCEMENT LEARNING USING RECURRENT SIAMESE NETWORKS

## ABSTRACT

It would be desirable for a reinforcement learning (RL)-based agent to learn behaviour by merely watching a demonstration. However, defining rewards that facilitate this goal within the RL paradigm remains a challenge. Here we address this problem with Siamese networks, trained to compute distances between observed behaviours and an agent's behaviours. We use a recurrent neural network (RNN)-based comparator model to learn such distances in space and time between motion clips while training an RL policy to minimize this distance. Through experimentation, we have also found that the inclusion of multi-task data and an additional image encoding loss helps enforce temporal consistency and improve policy learning. These two components appear to balance reward for matching a specific instance of a behaviour versus that behaviour in general. Furthermore, we focus here on a particularly challenging form of this problem where only a single expert demonstration is provided for a given task. We demonstrate our approach on simulated humanoid, dog and raptor agents in 2D and a 3D quadruped and humanoid. In these environments, we show that our method outperforms the state-of-the-art, Generative Adversarial Imitation from Observation (GAIfO) (i.e. Generative Adversarial Imitation Learning (GAIL) without access to actions) and Time-Contrastive Network (TCN).

## 1 INTRODUCTION

In nature, many intelligent beings (agents) can imitate their peers (experts) by watching them. In order to learn from observation alone, the agent must compare its own behavior to the expert's, mimicking their movements (Blakemore & Decety, 2001). While this process seems to come as second nature to humans and many animals, formulating a framework and metrics that can measure how different a expert's demonstration is from an agent's reenactment in this setting is challenging. While robots have access to their state information, humans and animals simply *observe* others performing tasks relying only upon visual perceptions of demonstrations, creating a mental representation of the target motion. In this work we ask: Can agents learn these representations in order to learn imitative policies from a single demonstration?

One of the core problems of imitation learning is how to align a demonstration in space and time with the agent's own state. To address this, the imitation framework has to learn a distance function between agent and expert. The distance function in our work makes use of positive and negative examples, including types of adversarial examples, similar to GAIL (Ho & Ermon, 2016) and GAIfO (Torabi et al., 2018b). These works require expert policies to generate large amount of demonstration data and GAIL need action information as well. These works train a discriminator to recognize in-distribution examples. In this work, we extend these techniques by learning distances between motions, using noisy visual data without action information, and using the distance function as reward signal to train RL policies. In Figure 1b an outline of our method for visual imitation is given. As we show in the paper, this new formulation can be extended to assist in training the distance function using multi-task data, which improves the model's accuracy and enables its reuse on different tasks. Additionally, while previous methods have focused on computing distances between single states, we construct a cost function that takes into account the demonstration ordering as well as the state using a recurrent Siamese network to learn smoother distances between motions.

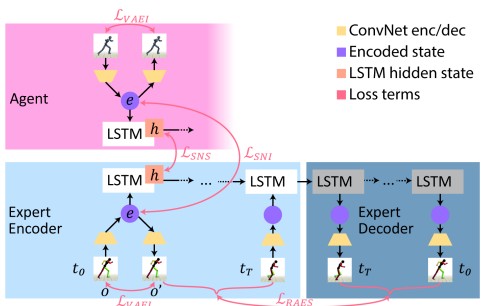 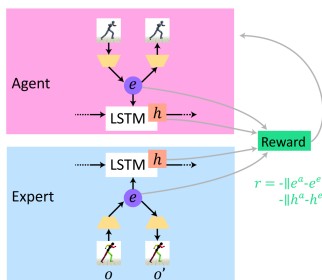

(a) Losses for training the encoders/decoders      (b) Reward generation for the agent

Figure 1: Overview of our method: We aim to learn a distance function (1a) and then use that distance function as a reward function for RL (1b). At the current timestep, observations (**o**) of the reference motion and the agent are encoded (**e**) and fed into LSTMs (leading to hidden states **h**). Fig. 1a shows how the reward model is trained using both Siamese and AE losses. There are: VAE reconstruction losses on static images ($\mathcal{L}_{VAEI}$), sequence-to-sequence AE losses ($\mathcal{L}_{RAES}$), one for the reference and one for the agent (which we do not show in pink to simplify the figure). There is a Siamese loss between encoded images ($\mathcal{L}_{SNI}$) and a Siamese loss that is computed between encoded states over time ($\mathcal{L}_{SNS}$). Fig. 1b shows how the reward is calculated **at every timestep**. Reward for the agent at every timestep consists of the distance between encoded images and encoded LSTM hidden states.

Our contribution, Visual Imitation with RL (VIRL), consists of proposing and exploring these forms of recurrent Siamese networks as a way to address a critical problem in defining the reward structure for imitation learning from video for deep RL agents. We accomplish this using simulated humanoid robots inhabiting a physics simulation environment and *for the challenging setting of imitation learning from a single expert demonstration*. Our approach enables us to train agents that can imitate many types of behaviours that include walking, running and jumping. We perform experiments for multiple simulated robots in both 2D and 3D, including recent Sim2Real quadruped robots and a humanoid with 38 degrees of freedom (DoF), *which is a particularly challenging problem domain*.

## 2 PRELIMINARIES

Here we provide a very brief review of some fundamental methods that are related to the new approach we present here. Reinforcement Learning (RL) is frequently formulated within the framework of Markov decision process (MDP) where at every time step $t$, the world (including the agent) exists in a state $s_t \in S$, where the agent is able to perform actions $a_t \in A$ and where states and actions are discrete. The action to take is determined according to a policy $\pi(a_t|s_t)$ which results in a new state $s_{t+1} \in S$ and reward $r_t = R(s_t, a_t, s_{t+1})$ according to the transition probability function $T(r_t, s_{t+1}|s_t, a_t)$. The policy is optimized to maximize the future discounted reward $\mathbb{E}_{r_0,...,r_T} \left[ \sum_{t=0}^{T} \gamma^t r_t \right]$, where $T$ is the max time horizon, and $\gamma$ is the discount factor, indicating the planning horizon length. The formulation above generalizes to continuous states and actions, which is the situation for the agents we consider in our work.

Imitation Learning is typically cast as the process of training a new policy to reproduce the behaviour of some expert policy. Behavioral cloning is a fundamental method for imitation learning. Given an expert policy $\pi_E$ possibly represented as a collection of trajectories $\tau = \langle (s_0, a_0), \ldots, (s_T, a_T) \rangle$ a new policy $\pi$ can be learned to match this trajectory using supervised learning and maximizing the expectation $\mathbb{E}_{\pi_E} \left[ \sum_{t=0}^{T} \log \pi(a_t|s_t, \theta_\pi) \right]$. While this simple method can work well, it often suffers from distribution mismatch issues leading to compounding errors as the learned policy deviates from the expert's behaviour (Ross et al., 2011b). Inverse reinforcement learning avoids this issue by extracting a reward function from observed optimal behaviour (Ng et al., 2000). In our approach, we learn a distance function that allows an agent to compare an observed behavior to its own current behavior to define its reward $r_t$ at a given time step. Our comparison is performed with respect to a reference activity but the comparison network can be trained across a collection of different behaviours. Further, we do not assume the example data to be optimal. See Appendix 7.2 for further discussion of the connections of our work to inverse reinforcement learning.

VAEs are a popular approach for learning lower-dimensional representations of a distribution (Kingma & Welling, 2014). A VAE consists of two parts, an encoder $q_\phi$, with parameters $\phi$ and a decoder $p_\psi$ with parameters $\psi$. The encoder maps inputs $\mathbf{x}$, to a latent encoding $\mathbf{z}$ and in turn the decoder transforms $\mathbf{z}$ back to the input space $p_\psi(\mathbf{x}||\mathbf{z})$. The model parameters for both $\phi$ and $\psi$ are trained jointly to maximize

$$\mathcal{L}_{VAE}(s, \phi, \psi) = -D_{KL}(q_\phi(\mathbf{z}||\mathbf{x})||p(\mathbf{z})) + \mathbb{E}_{q_\phi(\mathbf{z}||\mathbf{x})}[\log p_\psi(\mathbf{x}||\mathbf{z})], \tag{1}$$

where $D_{KL}$ is the Kullback-Leibler divergence, $p(\mathbf{z})$ is a prior distribution over the latent space. The encoder $q_\phi$, or inference model takes the form of a diagonal covariance multivariate Gaussian distribution $q_\phi = \mathcal{N}(\mu_\phi(\mathbf{x}), \sigma^2(\mathbf{x}))$, where the mean, $\mu_\phi(\mathbf{x})$ is typically given by a deep neural network.

Sequence to sequence models can be used to learn the conditional probability of one sequence given another $p(y_0, \ldots, y_{T'}|x_0, \ldots, x_T)$, where $\mathbf{x} = x_0, \ldots, x_T$ and $\mathbf{y} = y_0, \ldots, y_{T'}$ are sequences. Here we will use extensions of encoder-decoder structured, autoencoding recurrent neural networks which learn a latent representation $\mathbf{h}$ that compresses the information in $x_0, \ldots, x_T$. Our model for decoding the sequence $\mathbf{y}$ can then be written as

$$p(\mathbf{y}) = p(y_0|\mathbf{h}) \prod_{t=1}^{T} p(y_t|\{y_0, \ldots, y_{t-1}\}, \mathbf{h}). \tag{2}$$

This method has been used for learning compressed representations for transfer learning (Zhu et al., 2016) and 3D shape retrieval (Zhuang et al., 2015). In our case this type of autoencoding can help regularize our model, which has a primary goal of computing distances between sequences using a Siamese structured autoencoding RNN.

## 3 Visual Imitation with Reinforcement Learning

**High-level Overview** Our method is similar to other Imitation Learning frameworks like GAIfO in that we train a system to give the agent a reward depending on how closely it is imitating the expert. We interleave training between refining the reward generator with rollouts and using the reward generator to train the policy and gather more rollouts. The reward generator consists of several components and losses that are described in the following section but coarsely, observations of both the expert and agent are encoded with VAEs and LSTMs, to be later decoded in inverse order. Contrastive loss ("Siamese Network triplet loss") is used to maximize similarity between the encoding of similar frames/sequences and dissimilarity between incorrect frames and shuffled sequences[1](Hadsell et al., 2006). Once this system has been initialized, at every timestep a reward for the agent's policy is calculated as difference between the current encoded observation and also the difference of the sequence so far between expert and agent. In the following section, we first discuss how the encoder/decoder networks are trained, then how they generate reward for the agent, and finally which data augmentation techniques we used to make the system more robust.

**The Sequence Encoder/Decoder Networks** Figure 1a shows an outline of the system. A single convolutional network $\texttt{Conv}^e$ is used to transform observations (images) at time $t$ of the expert demonstration $\mathbf{o}_t^e$ to an encoding vector $e_t^e$. After the sequence of observations was passed through $\texttt{Conv}^e$ there is an encoded sequence $\langle e_0^e, \ldots, e_t^e \rangle$, this sequence is fed into the RNN $\texttt{LSTM}^e$ until a final encoding is produced $h_t^e$. This same process is performed for a copy of the RNN $\texttt{LSTM}^a$ producing $h_t^a$ for the agent $\mathbf{o}^a$. The final encoding of the expert is fed into a separate RNN $\texttt{LSTM}^{\hat{e}}$ which generates a series of decoded latent representations $\langle e_0^{\hat{e}}, \ldots, e_t^{\hat{e}} \rangle$ which are then decoded back to images with a deconvolutional network $\texttt{Deconv}^{\hat{e}}$. The same applied to the agent with RNN $\texttt{LSTM}^{\hat{a}}$, latent representations $\langle e_0^{\hat{a}}, \ldots, e_t^{\hat{a}} \rangle$, and deconvolutional network $\texttt{Deconv}^{\hat{a}}$, respectively.

**Loss Terms** The encoding of a single observation of either agent or expert at a given timestep is trained using the VAE loss $\mathcal{L}_{VAE}$ from Eq.1. A full sequence of observations of either agent or expert is encoded and then decoded back, and the $\texttt{LSTM}s$ are trained with the loss $\mathcal{L}_{RAES}$ from Eq.2. We found these frame- and sequence-autoencoders to improve latent space conditioning. A

---

[1]which is different from existing methods like GAIfO in that we enforce similarity over sequences, not just individual state transitions. This allows us to temporally align the demonstration with the agent.

frame-by-frame Siamese loss between $e_t^e$ of the expert and $e_t^a$ of the agent enforces individual frames to be encoded similarly. This Siamese Network image loss $\mathcal{L}_{SNI}$ is defined below in Eq.3. Lastly and primarily, a Siamese loss between a full encoded sequence of the expert $h_t^e$ and a sequence of the agent $h_t^a$ forces not just individual frames but the representation of whole sequences to match up if they are alike. This Siamese Network sequence loss $\mathcal{L}_{SNS}$ is also defined in Eq.3 since it uses the same formula, just expects a sequences instead of frames as input. The Siamese Network loss (both for images and sequences) is defined as:

$$\mathcal{L}_{SNX}(\mathbf{o}_i, \mathbf{o}_p, y; \phi) = y * ||f(\mathbf{o}_i; \phi) - f(\mathbf{o}_p; \phi)|| + ((1-y) * (\max(\rho - (||f(\mathbf{o}_i; \phi) - f(\mathbf{o}_n; \phi)||), 0))),$$
(3)

where $y \in [0, 1]$ is the indicator for positive/negative samples. When $y = 1$, the sample is positive and the distance between current observation $\mathbf{o}_i$ to positive sample $\mathbf{o}_p$ should be minimal. When $y = 0$, the sample is negative and the distance between $\mathbf{o}_i$ and negative example $\mathbf{o}_n$ should be maximal. This loss is computed over batches of data that are half positive examples and half negative. The margin $\rho$ is used as an attractor or anchor to pull the negative example output away from $\mathbf{o}_i$ and push values towards a $[0, 1]$ range. $f(\cdot)$ computes the output from the underlying network (i.e. `Conv` or `LSTM`). The data used to train the Siamese network is a combination of observation trajectories $\mathbf{O} = \langle \mathbf{o}_0, \dots, \mathbf{o}_T \rangle$ generated from simulating the agent in the environment and the expert demonstration. For our recurrent model the observations $\mathbf{O}_p, \mathbf{O}_n, \mathbf{O}_i$ are sequences. This combination of image-based and sequence-based losses assists in compressing the representation while ensuring intermediate representations remain informative. The combined loss to train the model on a *positive* pair of sequences ($y = 1$) is:

$$\mathcal{L}_{VIRL}(\mathbf{O}_i, \mathbf{O}_p, y; \phi, \psi, \omega, \rho) = \underbrace{\lambda_1 \mathcal{L}_{SNS}(\mathbf{O}_i, \mathbf{O}_p, y; \phi, \omega)}_{\text{Contrastive sequence loss}} + \underbrace{\lambda_2 \Big[ \frac{1}{T} \sum_{t=0}^{T} \mathcal{L}_{SNI}(\mathbf{O}_{i,t}, \mathbf{O}_{p,t}, y; \phi) \Big]}_{\text{Contrastive frame loss}} +$$

$$\underbrace{\lambda_3 [\mathcal{L}_{RAES}(\mathbf{O}_i; \phi, \psi, \omega, \rho) + \mathcal{L}_{RAES}(\mathbf{O}_p; \phi, \psi, \omega, \rho)]}_{\text{Recurrent autoencoder loss (full sequence)}} +$$

$$\underbrace{\lambda_4 \Big[ \frac{1}{T} \sum_{t=0}^{T} \big[ \mathcal{L}_{VAEI}(\mathbf{O}_{i,t}; \phi, \psi) + \mathcal{L}_{VAEI}(\mathbf{O}_{p,t}; \phi, \psi) \big] \Big]}_{\text{Variational autoencoder loss (individual frames)}}.$$
(4)

Where the relative weights of the different terms are $\lambda_{1:4} = \{0.7, 0.1, 0.1, 0.1\}$, the image encoder convnet is $\phi$, the image decoder $\psi$, the recurrent encoder $\omega$, and the recurrent decoder $\rho$.

**Reward Calculation** The model trained using the method described above is used to calculate the distance between two sequences of observations seen thus far up to time $t$ as $d(\mathbf{O}^e, \mathbf{O}^a; \phi, \omega) = ||\omega(\mathbf{o}_{0:t}^e; \phi) - \omega(\mathbf{o}_{0:t}^a; \phi)||$ and the reward as $r(\mathbf{o}_{0:t}^e, \mathbf{o}_{0:t}^a) = -d(\mathbf{O}^e, \mathbf{O}^a; \phi, \omega)$. This means at every timestep, the reward is computed as $r_t = ||h_t^e - h_t^a|| + ||e_t^e - e_t^a||$. This can be expanded to $r_t = ||\text{LSTM}^e(\text{CONV}^e(\mathbf{o}_{0:t}^a)) - \text{LSTM}^a(\text{CONV}^a(\mathbf{o}_{0:t}^a))|| + ||\text{Conv}^e(\mathbf{o}_t^e) - \text{Conv}^a(\mathbf{o}_t^a)||$ and is shown in Figure 1b. During RL training, we compute a distance given the sequence observed so far in the episode. This method allows us to train a distance function in the observations space where all we need to provide is labels that denote if two observations or sequences are similar or not.

**Training the Model** Details of the algorithm used to train the distance metric and policy are outlined in Algorithm 1. We consider a variation on the typical RL environment that produces 3 different outputs, two for the agent and 1 for the demonstration and no reward. The first is the internal robot pose, which we shall refer to as the state $s_t$. The second and third representation is the agent's rendered view, or observation $\mathbf{o}_t^a$ and the demonstration $\mathbf{o}_t^e$, shown in Figure 1b. The rendered views are used with the distance metric to compute the similarity between the agent and the demonstration. We learn the policy of our agents using RL and the Trust-Region Policy Optimization (TRPO) algorithm (Schulman et al., 2015) with a reward signal that is learned as discussed below.

**Unsupervised Data labelling** To construct *positive* and *negative* pairs for training, we make use of time information in a similar fashion to (Sermanet et al., 2017) and adversarial information similar to GAIL. Timing information is used where observations at similar times in the same sequence are often correlated, and observations at different times will likely have little similarity. We compute

these sequence pairs by altering one sequence and comparing this modified version to its original. Positive pairs are created by adding Gaussian noise with $\sigma = 0.05$ to the images in the sequence or swapping or duplicating random frames of the sequences. Negative pairs are created by shuffling, cropping or reversing one sequence. Additionally, we include *adversarial* pairs where positive pairs come from the same distribution, for example, two motions for the agent or two from the expert. Negative pairs then include one from the expert and one from the agent. More details are available in the supplementary document.

**Data Augmentation** We apply several data augmentation methods to produce additional data for training the distance metric. Using methods analogous to the cropping and warping methods popular in computer vision (He et al., 2015) we randomly *crop* sequences and randomly *warp* the demonstration timing. The *cropping* is performed by both initializing the agent to random poses from the demonstration motion and terminating episodes when the agent's head, hands or torso contact the ground. As the agent improves, the average length of each episode increases, and so to will the average length of the cropped window. The motion *warping* is done by replaying the demonstration motion at different speeds. Two additional methods influence the data distribution. The first method is Reference State Initialization (RSI) (Peng et al.,

---

**Algorithm 1 Learning Algorithm**

1: Initialize parameters $\theta_\pi, \theta_d, D \leftarrow \{\}$
2: **while** not done **do**
3:     **for** $i \in \{0, \dots, N\}$ **do**
4:         $\{s_t, \mathbf{o}_t^e, \mathbf{o}_t^a\} \leftarrow$ env.reset(), $\tau^i \leftarrow \{\}$
5:         **for** $t \in \{0, \dots, T\}$ **do**
6:             $a_t \leftarrow \pi(\cdot | s_t, \theta_\pi)$
7:             $\{s_{t+1}, \mathbf{o}_{t+1}^e, \mathbf{o}_{t+1}^a\} \leftarrow$ env.step($a_t$)
8:             $\tau_t^i \leftarrow \{s_t, \mathbf{o}_t^e, \mathbf{o}_t^a, a_t\}$
9:             $\{s_t, \mathbf{o}_t^e, \mathbf{o}_t^a\} \leftarrow \{s_{t+1}, \mathbf{o}_{t+1}^e, \mathbf{o}_{t+1}^a\}$
10:         **end for**
11:         $\mathbf{r}_{0:t}^i \leftarrow -d(\mathbf{o}_{0:t+1}^e, \mathbf{o}_{0:t+1}^a | \theta_d)$
12:     **end for**
13:     $D \leftarrow D \bigcup \{\tau^0, \dots, \tau^N, \}$
14:     Update $d(\cdot)$ parameters $\theta_d$ using $D$
15:     Update $\theta_\pi$ with $\{\{\tau^0, \mathbf{r}^0\}, \dots, \{\tau^N, \mathbf{r}^N\}\}$
16: **end while**

---

2018a), where the initial state of the agent and expert is randomly selected from the expert demonstration. With this property, the environment can also be thought of as a form of memory replay. The environment allows the agent to go back to random points in the demonstration as if replaying a remembered demonstration. The second is Early Episode Sequence Priority (EESP) where the probability a sequence $\mathbf{x}$ is cropped ending at $i$ is $p(i) = \frac{len(\mathbf{x}) - i}{\sum i}$, increasing the likelihood of starting earlier in the episode.

## 4 RELATED WORK

For the purposes of this work, we group existing imitation learning methods based on the type and quantity of data needed to learn. In the first tier, there is GAIL (Ho & Ermon, 2016) and related methods, which require access to expert policies, states and actions and require large quantities of expert data. In the second tier, the need for expert actions is relaxed in methods like GAIfO (Torabi et al., 2018b). In the third tier, the need for ground truth states is relaxed in favor of images which are easier to obtain in methods like T-REX and D-REX(Brown et al., 2019; 2020). These methods still require many examples of data from a policy trained on the agent in the same simulation with the same dynamics. Lastly, in the fourth tier, the need for multiple demonstrations and matching dynamics is relaxed in methods like TCN (Sermanet et al., 2018) and ours.

**Methods that require access to expert states and actions.** Generative Adversarial Imitation Learning or GAIL (Ho & Ermon, 2016), uses the well known Generative Adversarial Network (GAN) framework applied to learning an RL policy (Goodfellow et al., 2014). In GAIL, the GAN's discriminator is trained with positive examples from expert trajectories and negative examples from the current policy. However, using a discriminator is only one possible way of judging the distance between expert and agent and searching for good distance functions between states is an active research area (Abbeel & Ng, 2004; Argall et al., 2009; Finn et al., 2016; Brown et al., 2019). Given some vector of features, the goal of distance-based imitation learning is to find an optimal transformation of these features, such that in this transformed space, there exists a more meaningful distance between expert demonstrations and agent trajectories. Previous work has explored the area of state-based distance functions, but most rely on the availability of an expert policy to sample data (Ho & Ermon, 2016; Merel et al., 2017). In the section hereafter we demonstrate how VIRL learns a more stable distance-based reward over sequences of images (as opposed to states) and without access to actions or expert policies.

**Methods that don't require access to actions.** For learning from demonstrations (LfD) problems, the goal is to replicate the behaviour of an expert $\pi_E$. GAIfO (Torabi et al., 2018b) has been proposed as extension of GAIL that does not require actions. This and other recent works in this area require access to an expert policy to sample more states (Sun et al., 2019; Yang et al., 2019) . By comparison, our method only needs a single fixed demonstration. Other recent work uses behavioural cloning (BC) to learn an inverse dynamics model to estimate the actions used via maximum-likelihood estimation (Torabi et al., 2018a). Still, BC often needs many expert examples and tends to suffer from state distribution mismatch issues between the *expert* policy and *student* (Ross et al., 2011a).

Additional works learn implicit models of distance (Yu et al., 2018; Finn et al., 2017; Sermanet et al., 2017; Merel et al., 2017; Edwards et al., 2019; Sharma et al., 2019) they require large amounts of demonstration data and none of these explicitly learn a sequential model considering the demonstration timing. The work in (Wang et al., 2017; Li et al., 2017; Peng et al., 2018b) includes a more robust GAIL framework along with a new model to encode motions for few-shot imitation but they need access to an expert policy to sample data from. In this work, we train recurrent Siamese networks (Chopra et al., 2005) to learn more meaningful distances between videos. Other work uses state-only demonstration ranking to out-perform the demonstration data but requires many demonstrations and ranking information (Brown et al., 2019; 2020). We show results on more complex 3D tasks and additionally model distance in time, i.e. due to the embedding of the full sequence, our model can compute meaningful distances between agent and demonstration even if they are out of sync.

**Methods that work on images instead of states** Some works like Sermanet et al. (2017); Finn et al. (2017); Liu et al. (2017); Dwibedi et al. (2018), use image-based inputs instead of states but require many demonstrations. Further, these models only address spacial alignment (i.e. matching joint positions/orientations) but not temporal alignment (i.e. getting the sequence of motion correct rather than just the individual frames) between expert demonstration and agent motion like our recurrent sequence model does. Other works that perform imitation from only image-based information like (Pathak et al., 2018) do so between goal states.

**Methods that require few visual observations and allow for a different source environment** Time-Contrastive Networks (TCNs) (Sermanet et al., 2018) were proposed as a way to use a metric learning loss to embed *simultaneous viewpoints* of the same object. They use TCN embeddings as *features in the system state* which are provided to a reinforcement learning algorithm, specifically, PILQR (Chebotar et al., 2017) which combines model-based learning, linear time varying dynamics and model-free corrections. In contrast, our Siamese network-based approach is used to *learn the reward* for an arbitrary subsequent RL algorithm. Our method does not rely on multiple views and we use an RNN-based autoencoding approach to regularize the distance computations used for rewards generated by our models.

In summary, all existing methods either (a) require ground truth states and actions, (b) require access to states as opposed to images, (c) require large amounts of training data or require the expert to be trained in the same environment under the same dynamics, or (d) require the expert motion to be learned from spacial alignment alone, as opposed to spatiotemporal alignment. Our method requires none of these and therefore aims to provide a more generic solution.

## 5    RESULTS AND ANALYSIS

We use a collection of different simulation environments to validate VIRL's ability to train imitative agents. In these simulated robotics environment, the agent is learning to imitate a given reference demonstration. Each of these simulation environment provides a hard-coded reward function based on the robot's pose that is used to evaluate the policy quality independently. The demonstration $M$ the agent is learning to imitate is produced from a clip of mocap data. The mocap data is used to animate, kinematically, a second robot in the simulation. Frames from the simulation are captured and used as video input to train the distance metric. The images captured from the simulation are converted to grey-scale with $48 \times 48$ pixels. The policy instead received the state data, often as link distances and velocities relative to the robot's centre of mass (COM). These simulation environments are new and have all been updated to take motion capture data and produce view video data that can be used for training RL agents or generating data for computer vision tasks. The environment

includes challenging and dynamic tasks for humanoid, dog and raptor robots. Some example tasks are imitating running, jumping, trotting, and walking, shown in Figure 2 and Figure 3.

**2D Video Imitation Results** Our first experiments evaluate the method's ability to learn a complex cyclic motion for a simulated robots given a single motion demonstration, similar to (Peng & van de Panne, 2017), but instead using video. For each of these simulated robots VIRL is able to learn a robust gate even though it is only given noisy partial observations of a demonstration. Results for these environments can be found in Figure 2 (humanoid2d) and in Figure **??** (dog2d and raptor2d).

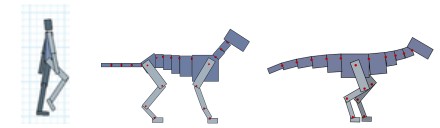

Figure 2: Frames from the humanoid2d, dog2d and raptor2d environments in our experiments.

**3D Robot Video Imitation** We train imitation policies from videos over a number of environments including two quadrupedal robots simulators used for Sim2Real research, the Laikago (Peng et al., 2020) and Pupper (Kau et al.). In these two quadruped simulators the environment is altered to produce additional video from a recorded demonstration of the robot performing a task. Additionally, we use environments with a simulated humanoid robot, the agent is learning to imitate a given reference motion of a walk, run, jump or zombie motion. A single motion demonstration is provided by the simulation environment as a cyclic motion. During learning, we can include additional data from all other tasks for the walking task this would be: walking-dynamic-speed, running, jogging, front-flips, back-flips, dancing, jumping, punching and kicking) that are only used to train the distance metric. We also include data from a modified version of the task that has a randomly generated speed modifier $\omega \in [0.5, 2.0]$ walking-dynamic-speed, which warps the demonstration timing. This additional data is used to provide a richer understanding of distances in space and time to the distance metric. The method is capable of learning policies that produce similar behaviour to the expert across a diverse set of tasks. We show example trajectories from the learned policies in Figure 3 and in the supplemental Video. It takes $5 - 7$ days to train each policy in these results on a 16 core machine with an Nvidia GTX1080 GPU.

**Algorithm Analysis and Comparison** In Figure 4a we show an evaluation of the learning capabilities and improvements of VIRL compared with two other methods that learn a distance function in state space, GAIfO (Torabi et al., 2018b) and a VAE trained to encode agent and reference observations and compute distances between those encodings, similar to Nair et al. (2018) and TCNs. We find that the VAE alone does not appear to model distances between states in a way that helps with RL, possibly due to the decoding complexity. Similarly, the

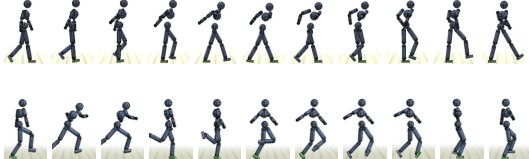

Figure 3: Rasterized frames of the agent's motion after training on humanoid3d walking and running. Additionally, a zombie walk and jumping policy can be found on the project website: https://sites.google.com/view/virl1. Also see Appendix Fig. 7.

GAIfO baseline produces very jerky motion or stands still, both of which are contained in the imitation distribution. Our full VIRL method considers the temporal structure of the data, learns faster and produces higher value policies.

In Figure 4b we compare the importance of adding the spatial VAE $||e_t^a - e_t^b||^2$ and temporal LSTM $||h_t^a - h_t^b||^2$ components of VIRL. Using the recurrent representation alone allows learning to progress quickly but can lead to difficulties informing the policy of how to best match the desired example. On the other hand, using only the encoding between single frames as is done with TCNs slows learning due to limited reward when the agent quickly becomes out-of-sync with the demonstration behaviour. We achieved the best results by combining the representations from these two models. This is shown for a completely different agent type (a 2d walking dog) and across many humanoid tasks in Figure 4(c-g). The use of multi-task data is not necessary but provides an improvement and was only used for the Walking task. Note that we experimented with using visual features as the state input for the policy as well; however, this resulted in poor policy quality.

**Ablation Analysis** We conduct ablation studies for learning policies for 3D humanoid control in Figure 5a and 5b. We compare the effects of data augmentation methods, network models and the use of additional data from other tasks (24 additional tasks like back-flips, see appendix 7.4). We compared using different length sequences for training, shorter (where the probability of the length decays

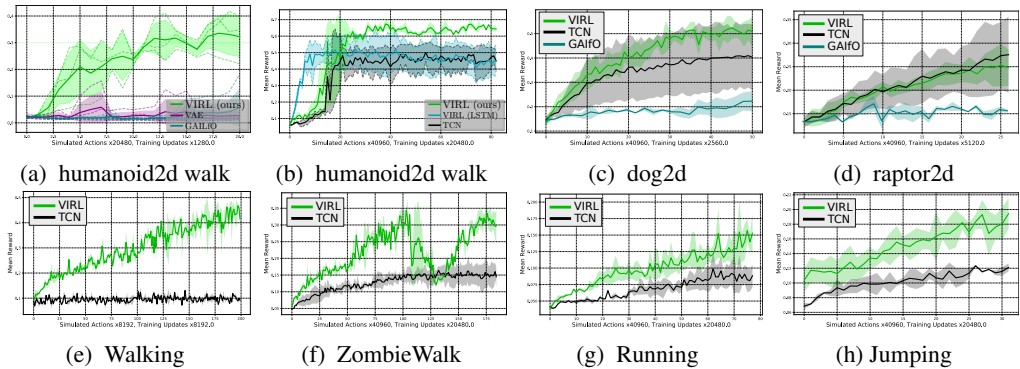

Figure 4: (a) Comparisons between VIRL, a simple VAE and GAIfO for the humanoid walking task. (b) Comparing our model with both an image VAE and and LSTM autoencoder (VIRL) with a model only having the LSTM autoencoder, versus a TCN. (c) Comparisons of VIRL with a TCN. In these plots, the large solid lines are the average performance of a collection of policy training simulations.

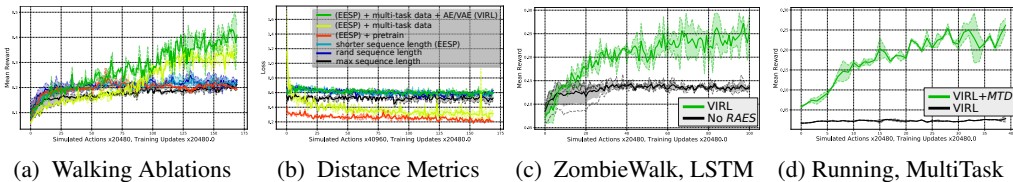

Figure 5: (a) Ablation analysis of VIRL on the Walking Task showing the mean reward over of the number of simulated actions. The legend is the same as (b) where we examine the impact on our loss under the different distance metrics resulting from the ablation analysis. We find that including multi-task data (only available for the humanoid3D) and both the VAE and recurrent AE losses provide the most performant models. (c) Ablating the recurrent autoencoder from VIRL dramatically impairs the ability to learn how to walk like a Zombie. (d) The use of multi-task training data helps learn better policies for running (away from Zombies if desired).

linearly), uniform random and max length available. For these more complex and challenging three dimensional humanoid (humanoid3d) control problems, the data augmentation methods, including EESP, increases average policy quality marginally. The use of multitask data Figure 5d and the additional recurrent sequence autoencoder (RSAE) greatly improves the methods ability to learn as observed in Figure 5c. As one can observe, our method performs better in this setting. Further analysis is available in the Appendix including additional comparison with TCNs in Figure 11(a-b) and using the 2D Raptor agent Figure 4d.

**Sim2Real for Quadreped Robots** We use VIRL to train policies for two simulated quadrapeds in Figure 6, that have been used for Sim2Real transfer. With these trained policies it is possible to transfer the VIRL policies trained from a single demonstration, to a real robot. The resultsing behaviours are availble

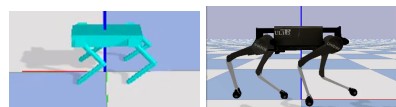

Figure 6: Pupper and Laikago Envs.

at: https://sites.google.com/view/virl1. We find that the Laikago environemnt is particularaly challenging to learn; however, we are able to learn good policies on the pupper in a day.

## 6 DISCUSSION AND CONCLUSION

In this work, we have created a new method for learning imitative policies from a single demonstration. The method uses a Siamese recurrent network to learn a distance function in both space and time. This distance function is trained on video data where the true state of the agent is noisily and partially observed. We use this to learn a reward function for training an RL policy. Using data from other motion styles and regularization terms, VIRL produces policies that demonstrate similar behaviour to the demonstration.

We believe VIRL will benefit from a more extensive collection of multi-task data and increased variation of each task. Additionally, if the distance metric confidence is available, this information could be used to reduce variance and overconfidence during policy optimization. We also believe that it is likely that learning a reward function while training adds additional variance to the policy gradient. This variance may indicate that the bias of off-policy methods could be preferred over the added variance of on-policy methods used here. Another approach may be to use partially observable RL that can learn a better value function model given a changing RNN-based reward function. Training the distance metric could benefit from additional regularization, such as constraining the kl-divergence between updates to reduce variance. Learning a sequence-based policy as well, given that the rewards are now not dependent on a single state observation is another area for future research that could improve performance.

We have compared our method to GAIfO, but we found GAIfO has limited temporal consistency. GAIfO led to learning jerky and overactive policies. The use of a recurrent discriminator for GAIfO may mitigate some of these issues and is left for future work. It is challenging to produce results better than the carefully manually crafted reward functions used by the RL simulation environments that include motion phase information in the observations (Peng et al., 2018a; 2017). However, we have shown that our method can compute distances in space and time and has faster initial learning. A combination of starting with our method and following with a manually crafted reward function, if true state information is available, could potentially lead to faster learning of high-quality policies. Still, as environments become increasingly more realistic and grow in complexity, we will need more robust methods to describe the desired behaviour we want from the agent. One might expect that the distance metric should be trained early and fast so that it quickly understands the difference between a good and bad demonstration. However, we have found that in this setting, learning too quickly can confuse the agent, as rewards can change, which can cause the agent to diverge off toward an unrecoverable policy space. In this setting, slower is better, as the distance metric may not yet be accurate. However, it may be locally or relatively reasonable, which is enough to learn a good policy. As learning continues, these two optimizations can converge together.

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

# 7 APPENDIX

This section includes additional details related to VIRL.

## 7.1 IMITATION LEARNING

Imitation learning is the process of training a new policy to reproduce the behaviour of some expert policy. BC is a fundamental method for imitation learning. Given an expert policy $\pi_E$ possibly represented as a collection of trajectories $\tau < (s_0, a_0), \ldots, (s_T, a_T) >$ a new policy $\pi$ can be learned to match these trajectory using supervised learning.

$$\max_\theta \mathbb{E}_{\pi_E} [\sum_{t=0}^{T} \log \pi(a_t | s_t, \theta_\pi)] \tag{5}$$

While this simple method can work well, it often suffers from distribution mismatch issues leading to compounding errors as the learned policy deviates from the expert's behaviour during test time.

## 7.2 INVERSE REINFORCEMENT LEARNING

Similar to BC, Inverse Reinforcement Learning (inverse reinforcement learning (IRL)) also learns to replicate some desired, potentially expert, behaviour. However, IRL uses the RL environment to learn a reward function that learns to tell the difference between the agent's behaviour and the example data. Here we describe maximal entropy IRL (Ziebart et al., 2008). Given an expert trajectory $\tau < (s_0, a_0), \ldots, (s_T, a_T) >$ a policy $\pi$ can be trained to produce similar trajectories by discovering a distance metric between the expert trajectory and trajectories produced by the policy $\pi$.

$$\max_{c \in C} \min_\pi (\mathbb{E}_\pi [c(s, a)] - H(\pi)) - \mathbb{E}_{\pi_E} [c(s, a)] \tag{6}$$

where $c$ is some learned cost function and $H(\pi)$ is a causal entropy term. $\pi_E$ is the expert policy that is represented by a collection of trajectories. IRL is searching for a cost function $c$ that is low for the expert $\pi_E$ and high for other policies. Then, a policy can be optimized by maximizing the reward function $r_t = -c(s_t, a_t)$.

## 7.3 PHASE-BASED IMITATION

If we consider a phase-based reward function $r = R(s, a, \phi)$ where $\phi$ indexes the time in the demonstration and $s$ and $a$ is the agent state and action. As the demonstration timing $\phi$, often controlled by the environment, and agent diverge, the agent receives less reward, even if it is visiting states that exist elsewhere in the demonstration. The issue of determining if an agent is displaying out-of-phase behaviour can be understood as trying to find the $\phi$ that would result in the highest reward $\phi' = \max_\phi R(s, a, \phi)$ and the distance $\phi' - \phi$ is an indicator of how far away in *time* or out-of-phase the agent is. This phase-independent form can be seen as a form of reward shaping. However, this naive description ignores the ordered property of demonstrations. What is needed is a metric that gives reward for behaviour that is in the proper order, independent of phase. This ordering motivates the creation of a recurrent distance metric that is designed to understand the context between two motions. For example, does this motion look like a walk, not, does this motion look precisely like that walk.

To encourage the agent to match any part of the expert behaviour, VIRL can be understood as decomposing the distance into two distances and learning Dynamic timme warping, by adding a type of temporal distance shown in green. To compute a time-independent distance we can find the state in the expert sequence that is closest to the agent's current state $\arg\min_{\hat{t} \in T} d(\mathbf{o}_{\hat{t}}^e, \mathbf{o}_t^a)$ and use it in the following distance measure

$$d^T(\mathbf{o}^e, \mathbf{o}^a, \hat{t}, t) = \ldots + d(\mathbf{o}_{\hat{t}-1}^e, \mathbf{o}_{t-1}^a) + d(\mathbf{o}_{\hat{t}}^e, \mathbf{o}_t^a) + d(\mathbf{o}_{\hat{t}+1}^e, \mathbf{o}_{t+1}^a) + \ldots \tag{7}$$

Using only a single state time-alignment may lead to the agent fixating on matching a particular state in the expert demonstration. To avoid this alignment issue, the neighbouring states are added to enforce sequential structure in the distance computation. This framework allows the agent to be rewarded for exhibiting behaviour that matches any part of the expert's demonstration. The better it

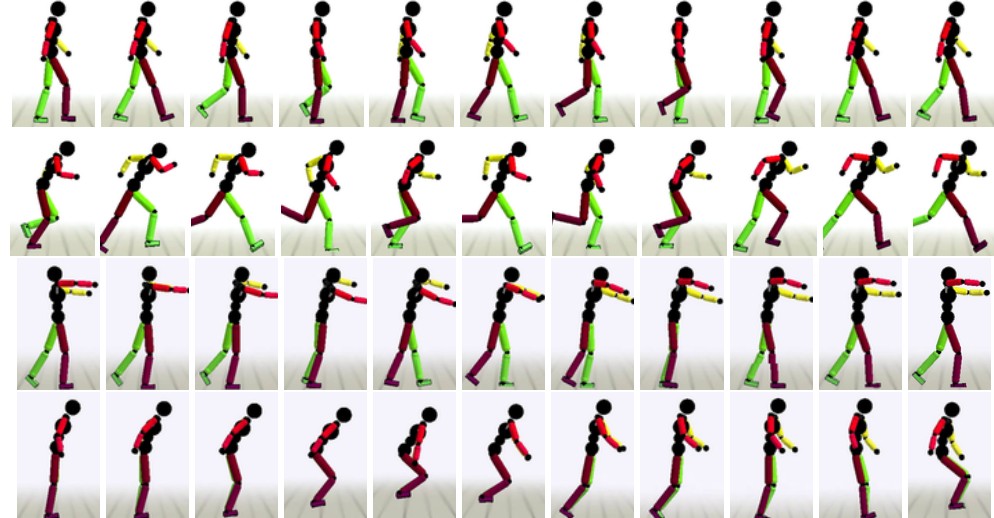

Figure 7: Rasterized frames of the imitation motions on humanoid3d walking (row 1), running (row 2), zombie (row 3) and jumping(row 4). https://sites.google.com/view/virl1

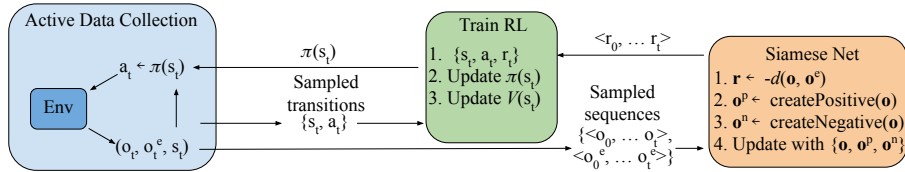

Figure 8: The flow of control for the learning system.

learns to match parts of the expert demonstration, the more reward it is given. The previous spatial distance will then help the agent learn to sync up the timing with the demonstration. Next, we describe how both distances can be learned together.

## 7.4 DATA

We are using the mocap data from the CMU Graphics Lab Motion Capture Database from 2002 (http://mocap.cs.cmu.edu/). To be thorough we provide the process at length. This data has been preprocessed to map the mocap markers to a human skeleton. Each recording contains the positions and orientations of the different joints of a human skeleton and can therefore directly be used to animate a humanoid mesh. This is a standard approach that has been widely used in prior literature like [Gleicher 1998, Rosales 2000, Lee 2002]. To be precise: at each mocap frame, the joints of a humanoid mesh model are set to the positions and orientations of their respective values in the recording. If a full humanoid mesh is not available, it is possible to add capsule mesh primitives between each recorded joint. This 3D mesh model is then rendered to an image through a 3rd person camera that follows the center of mass of the mesh at a fixed distance.

For the humanoid experiments, imitation data for 24 other tasks was used to help condition the distance metric learning process. These include motion clips for running, backflips, frontflips, dancing, punching, kicking and jumping along with the desired motion. The improvement due to these additional unsupervised training data generation mechanisms are shown in Figure 5a.

## 7.5 TRAINING DETAILS

The learning simulations are trained using graphics processing unit (GPU)s. The simulation is not only simulating the interaction physics of the world but also rendering the simulation scene to capture video observations. On average, it takes 3 days to execute a single training simulation. The process of rendering and copying the images from the GPU is one of the most expensive operations with VIRL.

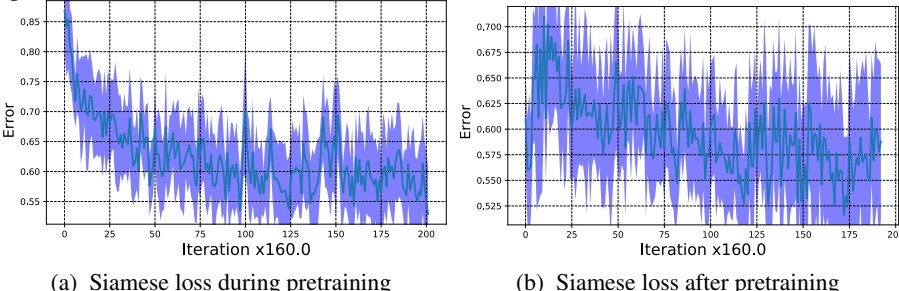

Figure 9: We use a Siamese autoencoding network structure that can provide a reward signal at every step to a reinforcement learning algorithm. For the Humanoid experiments here, the convolutional portion of our network includes 2 convolution layers of 8 filters with size $6 \times 6$ and stride $2 \times 2$, 16 filters of size $4 \times 4$ and stride $2 \times 2$. The features are then flattened and followed by two dense layers of 256 and 64 units. The majority of the network uses ReLU activations except for the last layer that uses a sigmoid activation. Dropout is used between convolutional layers. The RNN-based model uses a LSTM layer with 128 hidden units, followed by a dense layer of 64 units. The decoder model has the same structure in reverse with deconvolution in place of convolutional layers.

We collect 2048 samples between training rounds. The batch size for TRPO is 2048. The kl term is 0.5.

The simulation environment includes several different tasks that are represented by a collection of motion capture clips to imitate. These tasks come from the tasks created in DeepMimic (Peng et al., 2018a). We include all humanoid tasks in this dataset.

In Algorithm 1 we include an outline of the algorithm used for the method and a diagram in Figure 8. The simulation environment produces three types of observations, $s_{t+1}$ the agent's proprioceptive pose, $s_{t+1}^v$ the image observation of the agent and $m_{t+1}$ the image-based observation of the expert demonstration. The images are grayscale $64 \times 64$.

## 7.6 DISTANCE FUNCTION TRAINING

In Figure 10a, the learning curve for the Siamese RNN is shown during a pretraining phase. We can see the overfitting portion the occurs during RL training. This overfitting can lead to poor reward prediction during the early phase of training. In Figure 10b, we show the training curve for the recurrent Siamese network after starting training during RL. After an initial distribution adaptation, the model learns smoothly, considering that the training data used is continually changing as the RL agent explores.

|  |  |
|---|---|
| (a) Siamese loss during pretraining | (b) Siamese loss after pretraining |

Figure 10: Training losses for the Siamese distance metric.

It can be challenging to train a sequenced based distance function. One particular challenge is training the distance function to be accurate across the space of possible states. We found a good strategy was to focus on the earlier parts of the episode. When the model is not accurate on states it earlier in the episode, it may never learn how to get into good states later, even if the distance function understands those better. Therefore, when constructing batches to train the RNN on, we give a higher probability of starting earlier in episodes. We also give a higher probability of shorter sequences as a function of the average episode length. As the agent gets better average episodes length increase, so to will the randomly selected sequence windows.

### 7.7 DISTANCE FUNCTION USE

We find it helpful to *normalize* the distance metric outputs using $r = exp(r^2 * w_d)$ where $w_d = -5.0$ scales the filtering width. This normalization is a common method to convert distance-based rewards to be positive, which makes it easier to handle episodes that terminate early (Peng et al., 2018a;b; Peng & van de Panne, 2017). Early in training, the distance metric often produces large, noisy values. The RL method regularly tracks reward scaling statistics; the initial high variance data reduces the significance of better distance metric values produced later on by scaling them to small numbers. The improvement of using this normalized reward is shown in Figure 11a. In Figure 11b we compare to a few baseline methods. The *manual* version uses a carefully engineered reward function from (Peng et al., 2017).

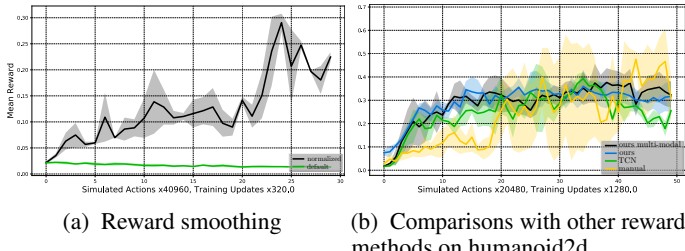

(a) Reward smoothing      (b) Comparisons with other reward methods on humanoid2d

Figure 11: Ablation analysis of VIRL. We find that training RL policies is sensitive to the size and distribution of rewards. We compare VIRL to a number of other simple baselines.

### 7.8 SEQUENCE ENCODING

Using the learned sequence encoder, we processed a collection of different motions to create a t-distributed Stochastic Neighbor Embedding (t-SNE) embedding of the encodings (Maaten & Hinton, 2008). In Figure 12b we plot motions both generated from the learned policy $\pi$ and the expert trajectories $\pi_E$. Overlaps in specific areas of the space for similar classes across learned $\pi$ and expert $\pi_E$ data indicate a well-formed distance metric that does not separate expert and agent examples. There is also a separation between motion classes in the data, and the cyclic nature of the walking cycle is visible. We also show how the choice in policy variance affects the RL jump task learning process in Figure 12a.

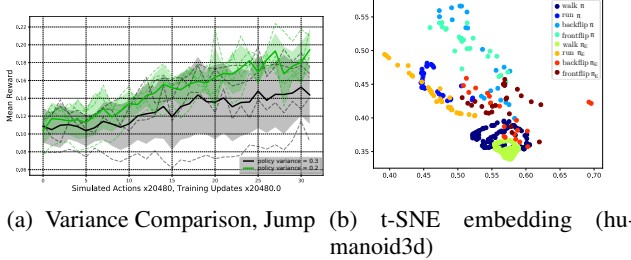

(a) Variance Comparison, Jump      (b) t-SNE embedding (humanoid3d)

Figure 12: Variance and t-SNE Visualization

### 7.9 POSITIVE AND NEGATIVE EXAMPLES

We use two methods to generate positive and negative examples. The first method is similar to TCN, where we can assume that sequences that overlap more in time are more similar. For each episode, two sequences are generated, one for the agent and one for the imitation motion. Here we list the methods used to alter sequences for positive pairs.

1. Adding Gaussian noise to each state in the sequence (mean = 0 and variance = 0.02)
2. Out of sync versions where the first state is removed from the first and the last ones from the second sequence
3. Duplicating the first state in either sequence
4. Duplicating the last state in either sequence

We alter sequences for negative pairs by

1. Reversing the ordering of the second sequence in the pair.

2. Randomly picking a state out of the second sequence and replicating it to be as long as the first sequence.

3. Randomly shuffling one sequence.

4. Randomly shuffling both sequences.

The second method we use to create positive and negative examples is by including data for additional classes of motion. These classes denote different task types. For the humanoid3d environment, we generate data for walking-dynamic-speed, running, back-flipping and front-flipping. Pairs from the same tasks are labelled as positive, and pairs from different classes are negative.

## 7.10 RL ALGORITHM ANALYSIS

It is not clear which RL algorithm may work best for this type of imitation problem. A number of RL algorithms were evaluated on the humanoid2d environment Figure 13a. Surprisingly, TRPO (Schulman et al., 2015) did not work well in this framework, considering it has a controlled policy gradient step, we thought it would reduce the overall variance. We found that Deep Deterministic Policy Gradient (DDPG) (Lillicrap et al., 2015) worked rather well. This result could be related to having a changing reward function, in that if the changing rewards are considered off-policy data, it can be easier to learn. This can be seen in Figure 13b where DDPG is best at estimating the future discounted rewards in the environment. We also tried Continuous Actor-Critic Learning-Automaton (CACLA) (Van Hasselt, 2012) and Proximal Policy Optimization (PPO) (Schulman et al., 2017); we found that PPO did not work particularly well on this task; this could also be related to added variance.

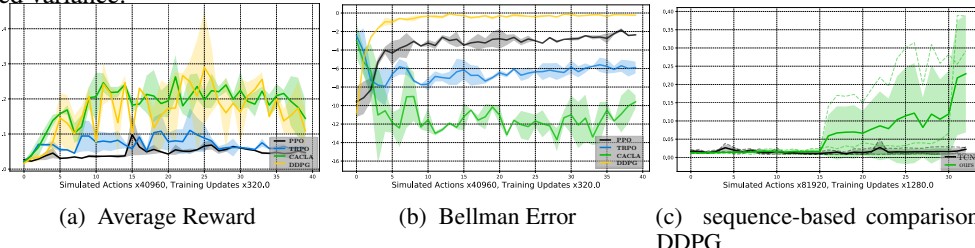

(a) Average Reward     (b) Bellman Error     (c) sequence-based comparison DDPG

Figure 13: RL algorithm comparison on humanoid2d environment.

