# OpenReview forum: "Visual Imitation with Reinforcement Learning using Recurrent Siamese Networks"
_ICLR.cc/2021/Conference — Reject_

### Official Review · AnonReviewer3 · 2020-10-19
**Review of "Visual Imitation with Reinforcement Learning using Recurrent Siamese Networks"**

**Rating:** 4
**Confidence:** 4

**Review:**

**Summary**: This paper studies the problem of visual imitation learning: given a video of an expert demonstration, take actions to reproduce that same behavior. The proposed method learns a distance metric on videos and uses that distance metric as a reward function for RL. Experiments show that this method does recover reasonable behaviors across a range of simulated robotic tasks.  Compared with prior methods, the main contribution of this work is that the distance metric is parametrized and trained as a siamese network.

**Novelty and Significance**: While the exact method seems novel, it is very similar to a number of prior methods, most notably GAIL. At a high-level the main difference is that this paper uses a siamese network to parametrize the discriminator, and employs a few types of data augmentation. If the experiments had shown that this architectural choice made a significant improvement in performance, and was useful inside a range of imitation learning frameworks (e.g., GAIL, AIRL, and Value Duce), then I think it'd represent a significant contribution. As is, the paper has not convincingly shown that this architectural choice is critical for significantly improved performance.


**Experiments:**
* I found it challenging to assess the experimental results without any quantitative comparisons with baselines (besides TCN). I'd highly recommend comparing against recent imitation learning methods. For example, some reasonable baselines would be
  * GAIL or InfoGAIL
  * Zero-shot visual imitation
  * Imitating latent policies from observation
  * Generative adversarial imitation from observation (Though this method is discussed, it's not compared against in the figures.)
  * Learning robust rewards with adversarial inverse reinforcement learning
  * Discriminator-actor-critic: Addressing sample inefficiency and reward bias in adversarial imitation learning
  * Imitation learning via off-policy distribution matching
* The videos of the humanoid imitation behavior on the accompanying website actually look quite poor. The videos for other domains seemed to have been rendered incorrectly.
* I found it somewhat irksome that the method is entitled "visual imitation" but the learned policy doesn't take visual inputs ("Note that we experimented with using visual features as the state input for the policy as well; however, this resulted in poor policy quality.")



**Clarity:**
* The introduction makes imitation learning sound like a new problem. I'd recommend clarifying the relationship with prior work earlier in the introduction.
* I found Fig 1 hard to follow, largely because it's unclear what the method is supposed to produce. One idea is to explicitly say something like: "We aim to learn a distance function (left) and then use that distance function as a reward function for RL (right). The distance function is learned by ..."
* I found Eq 4 hard to parse. It might be helpful to label each term (using \underbrace{}) with its semantic meaning.
* Reward Calculation: One thing that's unclear here is which trajectories are used for computing the reward function. I'm guessing that the agent's trajectory is compared to an expert trajectory, but I don't think this is ever stated explicitly. If so, it's unclear how to compute the reward function when multiple expert trajectories are given.


Overall, I give this paper a score of 4/10, primarily because of the lack of discussion and comparison with prior work. I would consider increasing my score if the paper compared against recent visual imitation learning methods (see above). The proposed method also seems quite similar to GAIL; if possible, it'd be great to formally explore the connections between these two methods. Finally, I'd recommend focusing the related work section to only discuss the most related works (see list above), but to discuss the exact differences between these prior methods and the proposed method in more depth.

**Questions for discussion**:
* Precisely, what are the differences between this method and GAIL? (I want to make sure I didn't miss an important difference)
* For the experiments in "3D Robot Video Imitation", is the expert demonstration provided as an RGB video, a mocap trace, or something else?
* Where is the comparison with GAIfO shown? ("We have compared our method to GAIfO")
* "that takes into account demonstration ordering" -- In a Markovian environment with a Markovian expert, shouldn't comparisons based on state-action pairs be sufficient? Is the assumption that the expert's motion is not Markovian?
* "Mutual information loss (”Siamese Network triplet loss”)" -- Eq 3 doesn't look like a mutual information objective. Can you explain precisely how maximizing Eq 3 leads to maximizing mutual information?


**Minor comments:**
* Abstract: No need to define "GAIfO" as an acronym in the abstract. Instead, just use "GAIL without access to actions."
* "In nature ... their movements" -- Please add citations.
* "formulating ... is challenging" -- How is this different from the large body of prior work on imitation/apprenticeship learning?
* "The fundamental problem ..." -- This is only true for *trajectory*-based imitation learning, not *transition*-based imitation learning (e.g., BC, GAIL, AIRL).
* "re-use" -> "reuse"
* "Additionally, While" -> "Additionally, while"
* "Including, ..." This sentence is missing a subject.
* "Markov Decision Processes" -> "Markov decision processes"
* When defining a trajectory, use \langle and \rangle.
* "it often suffers from distribution mismatch" -- Cite DAGGER, or something similar.
* "behaviour Ng" -- Use \citep{} for references that are not used as nouns.
* Eq 2: Can you define the distribution over $h$? (I assume it will be p(h | x)).
* Eq 3: Perhaps cite prior methods that use this contrastive margin loss (e.g., FaceNet)
* Define VIRL = "Visual Imitation with RL" in S3. It took me awhile to figure out what VIRL was when it was first mentioned in S4.
* "is an active research area" -- I agree, but using citations from '04 and '09 doesn't make this area sound particularly "recent."
* "the goal is to" -- The goal of what? Of all "good distance functions"?
* "state-based metrics ... image based inputs" -- Aren't image-based metrics a special case of state-based metrics? For example, the [Ho & Ermon] citation for state-based metrics can be applied to images.
* "Additional works..., none of these ..." -> "While additional works ..., none of these ..."

----------------------------
**Update after author response**: Thanks to the authors for answering my questions and for incorporating feedback into the paper. Through discussion, I think we got to the crux of the method: distance functions seem to work better than classifiers for imitation learning. I think this is a really neat observation, and potentially quite important; the experiments definitely support this hypothesis. That said, I don't think the paper goes far enough in exploring this hypothesis, either experimentally or theoretically. There are a number of confounding variables, such as data, loss function, and architecture, which each will need to be accounted for. I therefore stand by my previous vote (4) to reject the paper. With more thorough experiments and ablations, I think this will make a fantastic submission to a future conference.

---

> ### Author Response · Authors · 2020-11-19
> **Response 2/2**
>
> Most of your minor comments have been changed accordingly in the document. Thank you for these suggestions. Regarding your questions:
>
> - `"formulating ... is challenging" -- How is this different from the large body of prior work on imitation/apprenticeship learning?` - This is a problem that is often addressed in the imitation learning literature. We hope that we’re making this clear later on in the 3rd paragraph of the Related Works section (starts with “Searching for good distance functions between states is an active research area…”). We think that besides a large body of work, this remains a hard problem.
> - `“Aren't image-based metrics a special case of state-based metrics?”` - We think this _can_ work but isn’t guaranteed to because the lower observability in images could potentially be detrimental to certain models.
>
> For your convenience, here's the link to the side-by-side comparison with the previous iteration of the paper: https://openreview.net/revisions/compare?id=MBdafA3G9k&left=Nqvt8Vplgz&right=cHuGdDlXGd

---

> > ### Comment · AnonReviewer3 · 2020-11-21
> > **Updated review after reading revised paper**
> >
> > Thanks for the detailed response! I have gone through the revised paper and above comments.
> >
> > > Relationship with GAIL and GAILfO
> > Thanks for the detailed discussion. Indeed, the experimental results comparing with GAILfO are quite impressive! Now, the question is *why* does VIRL do so much better than GAILfO, given than their objectives are so similar? Is it because of the neural network architecture used for the classifier/distance function? Is there some important difference in their objective functions?
> >
> > > “The videos for other domains seemed to have been rendered incorrectly”?
> > In the videos of the Pupper and Liakago I see 8 copies of the robot in each frame. In the Dog2D and Raptor2D videos I see ~20 copies of the robot in each frame.
> >
> > > Comparison tool
> > Thanks for pointing out this feature! I didn't know about this.

---

> > > ### Author Response · Authors · 2020-11-23
> > > **Further details**
> > >
> > > Re: why does VIRL do so much better than GAILfO
> > >
> > > In general GAN-based methods are difficult to train well. We initially looked into methods based on GANs but discovered that for them to work well a significant amount of data is needed to train a good discriminator. It is for this reason we explored less data-hungry methods based on Siamese networks that we have found to work much better for this complex task where we only have one positive example. This can be seen in Figure 4 where TCN and GAILfO share the same network architecture but TCN performs much better. Our method then further improves on TCN.
> > >
> > > Re: Rendering
> > >
> > > Our apologies for this confusion. These videos show multiple rollouts concatenated together into a row of videos. The Pupper and Laikago examples show 8 different rollouts next to each other and 20 for the Dog2D and Raptor2D. In this form, we can get a better understanding of the overall behavior of the learned agent.

---

> > > > ### Comment · AnonReviewer3 · 2020-11-24
> > > > **Response**
> > > >
> > > > **Rendering** -- Thanks for clarifying; that makes sense!
> > > >
> > > > **VIRL vs GAILfO** -- Why is learning a distance function easier than learning a discriminator? Are they trained on different data? Is the slight different in objective functions important?

---

> > > > > ### Author Response · Authors · 2020-11-25
> > > > > **Follow up**
> > > > >
> > > > > ViRL vs GAIfO
> > > > >
> > > > > ViRL actually learns a conditional distance function $d(O^e,O^a)$. This means that a different demonstration can be provided to VIRL and the VIRL agent will learn how to imitate this new demonstration. This is explored in Figure 5 (a) where we show we can improve the performance of VIRL using data from other tasks to help train the distance function. With this conditioning, VIRL is able to train its distance function better as it allows the distance function to be shared across tasks.

---

> ### Author Response · Authors · 2020-11-19
> **Response 1/2: Authors appreciate very detailed review, added all minor fixes, reworked Rel. Works section and hope to clarify difference to GAIL/GAIfO**
>
> Dear Reviewer 3,
>
> Thank you so much for your time and effort!
>
> Your main concern seems to be related to a perception that our method is too similar to GAIL and an impression that our method doesn’t show significant improvement over SotA techniques like GAIL or similar.
>
> We would like to make some very important clarifications. Our approach is substantially different from GAIL. GAIL requires that the expert demonstration contains actions. Our method does not rely on having access to the actions used in the reference example. For a simple robot, this can be obtained by recording the instructions, but for a human, this can only be done from motion-tracking, without access to the ground-truth forces that the human applied to each joint. Since we want our method to address this and be more generally usable than GAIL, we do **not** require actions. A GAIL-like method that also works without actions is Generative Adversarial Imitation from Observation (GAIfO). We implemented this method and demonstrate in Fig. 4a a significant improvement of our method over the GAIfO baseline (GAIfO might be hard to see because it is hugging the x-axis).
>
> We show that our method outperforms current SotA in imitation learning from observation (without access to actions) by a notable margin. We hope this addressed the aforementioned main concern of the reviewer. If there should be other major concerns, we are happy to address them.
>
> With regards to your other comments:
>
> - **Experiments - Q1**  was hopefully addressed in the answer above.
> - **Experiments - Q2** - “The videos for other domains seemed to have been rendered incorrectly”? These videos appear to be rendered correctly. If you are referring to the additional agent in the simulation this is a visualization of the demonstration to help the view see the difference between the learned motion and the demonstration.
> - **Experiments - Q3** - The title part “Visual Imitation” comes from our distance function, which is trained from images. We do not want to train the policy from the privileged 3rd person image information that an agent in the real world will not have access to. The agent will have access to its internal sensors, so we use that for the input to the policy.
> - **Clarity - Q1** - We have rephrased part of our introduction to tone down this statement.
> - **Clarity - Q2** - We have updated the paper to include this.
> - **Clarity - Q3** - Thanks, we have updated Eq.4 accordingly.
> - **Clarity - Q4** - Only a single expert trajectory is ever given. That’s one of the main contributions of this work. We train the frame and sequence encoders with other trajectories too but at test time, the encoders are frozen and only used to compare the agent trajectory to the one expert demonstration.
>
> Re: Questions for discussion
>
> **Q1 (differences between this method and GAIL)**
> A1: We provided a more general comment on the problem setting and here we outline the general differences between our method and GAIL here. GAIL requires, access to an expert policy so that expert actions and true state information can be sampled at length from the environment to learn how to discriminate between the expert and agent. Our method is targeting a more natural and difficult imitation learning system where none of this information is available. The agent only has access to a single demonstration from another agent that can have different dynamics.
>
> **Q2: where do expert demonstrations for the 3d robot come from?**
> A2:  For the Laikago robot, the expert demonstration comes from a kinematic motion clip of a dog that is part of the CMU database. The mocap data used for the Stanford Quadruped Pupper robot comes from recording motion on the real robot which was remote-controlled to walk via a bezier gait.
>
> **Q3: Where is the comparison with GAIfO shown?**
> A3: We include a comparison to GAIfO in Figure4 (a) for the humanoid2d environment.
>
> **Q4: with a Markovian expert, shouldn't comparisons based on state-action pairs be sufficient?**
> A4: This is true, however, we only have access to partial information in the form of image information of the demonstration. In this setting, the expert data can be viewed as non-Markovian do to only having access to partial information
>
> **Q5: “Can you explain precisely how maximizing Eq 3 leads to maximizing mutual information?”**
> A5: We apologize for the misnomer. The correct term was “contrastive loss”, not “mutual information loss”. We have updated this in the paper.
>
> reply continued in next post...

---

### Official Review · AnonReviewer2 · 2020-10-27
**Limited contributions, needs some polish**

**Rating:** 4
**Confidence:** 3

**Review:**

### Paper summary

This paper proposes using a recurrent siamese network to learn distance functions between observed behaviours and agent behaviours, and the use of this distance for reinforcement learning. Experiments on a set of simulated walking tasks show that this approach works reasonably well.

Unfortunately, I am recommending rejection, as I do not think the contributions of this work are significant enough to warrant publication at ICLR. Siamese recurrent networks are already well established techniques for distance learning ([Chopra et al. 05](http://yann.lecun.com/exdb/publis/pdf/chopra-05.pdf), [Pei et al. 16](https://arxiv.org/pdf/1603.04713.pdf)), so it seems the primary contribution here is a method of artificially adding noise to help distinguish expert and demonstrated behaviours in training, an idea already explored to a large extent by TCNs and D-REX.

### Pros
- The proposed approach performs well when compared against GAILfO and learns better distance functions than a TCN.
- The results look good, and I really like the zombie walk.

### Cons

- The ideas in this work are very close to T-Rex [Brown et al. 19a](https://arxiv.org/abs/1904.06387) and D-Rex [Brown et al. 19b](https://arxiv.org/pdf/1907.03976.pdf) which uses a pairwise trajectory ranking formulation, which essentially learns the distance between trajectories, much like a siamese network. Brown et al. 19b also explore artificial trajectory ranking by adding noise to trajectories.
- The proposed approach relies on a number of heuristics and connected neural components and seems to perform well in walking tasks, but makes no real theoretical contribution.

### General comments and queries

- The paper seems to have been quite rushed and in need of some refinement when it comes to presentation. Specifically, please use a spell checker - there are numerous typos where letters are missing from words alongside spelling errors (eg. Advasarial,  quadreped, etc...), and check figure sizes to make sure legends are legible. The accompanying website is also full of broken links and unfilled template information.
- Some sweeping statements need toning down. Eg. in the introduction it is claimed that *The fundamental problem of imitation learning is how to align a demonstration in space and time with the agent’s own state.* which is arguable, I've always thought the fundamental problem in imitation learning is determining which part of a demonstration is essential and which is ancillary.
- I'd recommend a discussion and comparison with [D-REX Brown et al. 19b](https://arxiv.org/pdf/1907.03976.pdf)
- Equation 4 - how were the weights for these heuristic losses chosen?

---

> ### Author Response · Authors · 2020-11-19
> **Clarified difference to T/D-REX and contribution in reworked Rel. Works section**
>
> Dear Reviewer 2,
>
> Thank you for your time and effort!
>
> We would like to address the main concern of the reviewer first: the perceived limited contribution of this work and the overlap with existing work.
> The contribution of this work is to use the distance function that was learned by a Recurrent Siamese Network as the reward signal for training a policy and, crucially, providing an experimental framework that is able to do so from a **single** expert demonstration and no other information.
>
> **Q: Similarity of this work with T-REX and D-REX.**
> A: These two papers are able to perform imitation using a few, 10s to 100s, of examples. Also, this method requires supervised ranking information of the demonstration. Our method only needs **one single demonstration** of the behavior.
>
> We would also like to point out that in both the T- and D-REX papers, the authors assume that an expert policy can be trained in the environment through RL to sample the demonstrations needed for imitation. We make no such assumption. We don’t even require access to the **actions** of the experts. All we need is a video with the observations (images). Since we don’t rely on expert policies being trained in the simulation beforehand, we can also solve tasks that are easy to demonstrate but hard to learn through pure RL like full 3D humanoid locomotion.
>
> We hope that this addresses the main issue the reviewer had and allows them to reevaluate the paper. If there should be any other major concerns, we are happy to address them.
>
> With respect to the other, minor concerns:
>
> - We acknowledge that there were indeed a large number of grammatical mistakes in the paper. We apologize and we hope that in the new version, we have corrected these.
> - The sweeping statement in the introduction has been toned down to better represent the work in the paper.
> - We have added a few sentences discussing D-REX and T-REX to the related works section.
> - “Equation 4 - how were the weights for these heuristic losses chosen?” The weights were selected from a hyper parameter search over reasonable values.
>
> For your convenience, here's the link to the side-by-side comparison with the previous iteration of the paper: https://openreview.net/revisions/compare?id=MBdafA3G9k&left=Nqvt8Vplgz&right=cHuGdDlXGd

---

> > ### Comment · AnonReviewer2 · 2020-11-20
> > **Thanks for the rapid response, I still think there are strong similarities with T-REX and D-REX being missed**
> >
> > Thank you for engaging with the process, and taking my comments on board.
> >
> > Unfortunately,  I still don't think it is fair to disregard D-REX, which to me is fundamentally doing the same thing as this work, using pairwise behaviour ranking with artificial noise injection to generate ranking data. This implicitly learns a distance function that can be used as a reward. D-REX could still learn from a single demonstration, as it uses artificial trajectory ranking through noise injection.  Moreover, while D-REX is proposed within a policy search setting, the same algorithm can be directly applied to raw observations alone - it is after all just a ranking loss.
> >
> > There are certainly architectural and application differences, but I think the fundamental idea here is the same, and can't be disregarded so easily.

---

> > > ### Author Response · Authors · 2020-11-23
> > > **Further discussion**
> > >
> > > Dear Reviewer 2,
> > >
> > > Thank you for your quick response.
> > >
> > > Both D-REX and our method lead to successful imitation learning. However, there are differences between both methods in assumptions, algorithms, and experiments, that we think are sufficiently novel compared to D-REX:
> > >
> > > - T-Rex requires supervised information in the form of rankings among a collection of demonstrations. Given this ranking requirement, it would appear impossible T-Rex to recover the expert reward function with only a single demonstration. VIRL does not require this information and is capable of learning a distance function between sequences using a single demonstration.
> > > - D-REX does only need a single demonstration but requires that this demonstration includes action information to perform behavior cloning. In addition, D-REX requires access to the environment to generate additional noisy trajectories from the expert to create a collection of ranked trajectories.
> > > - VIRL does not require this information and is capable of learning a distance function between sequences using a single observation-only demonstration.
> > >
> > >
> > > - D-REX assumes that there’s a ground-truth reward function (R*), a reward function learned through IRL from the expert (R^), and that the real reward function can be approximated as the expert reward function plus a small error (R*=R^+e, at each timestep). This works well if the source and target environment follow the same dynamics (e.g. bodies move at the same speed). But if for example, the expert moves faster than the agent can move, this causes misalignment and the agent can’t learn a meaningful reward function.
> > > - In comparison, our method makes no assumption of any relationship between the expert’s reward and the ground truth reward. We only take the distance between the expert’s sequence and the agent’s sequence as well as the distance between the expert’s current embedded observation and the agent’s current embedded observation as the reward. All other methodological and implementation differences aside, we think this is the biggest difference between our method and D-REX: We compare sequences and frames, D-REX compares individual frames. Therefore D-REX can not handle temporal alignment.
> > > - This is a side-effect of point 1, but just to be clear: D-REX positions itself as an Inverse Reinforcement Learning method, i.e. something that can learn the reward function of the expert. In our work, we do not need to recover the expert reward function. In its place, we aim to learn a distance function between sequences and frames.
> > > - Fundamentally, our method doesn’t rank anything and doesn’t require noise injection. We learn a sequence and frame encoder function and that is all. D-REX, however, learns the entire reward function. We hope it is clear how these two things are different.
> > > - We think it is great that D-REX _can_ also work with only a single demonstration but the original paper doesn’t show any evidence to that extent, while we do. For the camera-ready version of this paper we can add a comparison of our method to D-REX.

---

> > > > ### Comment · AnonReviewer2 · 2020-11-24
> > > > **Further discussion**
> > > >
> > > > Thanks for continuing to indulge me and apologies if I sound like a broken record.
> > > >
> > > > I guess the main reason I keep returning to similarities with D-REX is that a Siamese loss is a ranking loss, even if this paper isn't interpreting it this way, and the distance function learned between encoded states can still be though of as a reward function, particularly since the distance is being used for imitation.
> > > >
> > > > Moreover, from my understanding of the paper, D-Rex does rank sequences as a whole - the ranking loss is over the cumulative reward over entire trajectories.
> > > >
> > > > I do acknowledge that the inclusion of the recurrency in the latent state means that temporal information is better exploited here, so understandably VIRL would learn a better distance/reward function, but I don't think this is a particularly significant finding.

---

> > > > > ### Author Response · Authors · 2020-11-25
> > > > > **Extended Discussion**
> > > > >
> > > > > It is np at all. We appreciate the discussion on the differences and similarities between VIRL and D-Rex.
> > > > >
> > > > > There are indeed conceptual similarities between the methods. In the paper, we do motivate our method via IRL and any method that is performing some type of imitation is going to need a function that can measure if the produced behaviour is similar to the desired behaviour. There are many papers that fit this description. However, each method works under a different set of assumptions and requirements and as a result has different guarantees on the produced policy and in what settings the method can be successful. Here we provide a few more detailed comments on the properties and differences of our method.
> > > > >
> > > > > - ViRL actually learns a conditional distance function $d(O^e,O^a)$.  This means that a different demonstration can be provided to VIRL and the VIRL agent will learn how to imitate this new demonstration. This is explored in Figure 5 (a) where we show we can improve the performance of VIRL using data from other tasks to help train the distance function.
> > > > >
> > > > > - In our work, we found the introduction of the recurrent loss and structure was one of the most important parts. As shown in Figure 4, without it many of the tasks can not be learned at all. The use of the recurrent distance allows VIRL to reward the agent for performing behaviour with a similar style to the demonstration without having to match the demonstration exactly. For example, when learning to walk VIRL can reward the agent for first learning to stand as that behaviour exists inside the distribution of walking.
> > > > >
> > > > > We believe these to be important points as they have made the method work much better in practice than other methods in this setting.

---

### Official Review · AnonReviewer1 · 2020-10-28
**Interesting method, but experiments section is lacking**

**Rating:** 5
**Confidence:** 4

**Review:**

SUMMARY:
The authors propose an extension of recent imitation from observation techniques called VIRL that explicitly incorporates Siamese network and LSTM encodings in order to attempt to better overcome some of the challenges of imitation learning from visual observations. They evaluate their algorithm in several simulation domains and find reasonable results.


STRENGTHS:
	(S1) The paper considers an important and timely problem in trying to improve upon the state-of-the-art in imitation learning from visual observations.
	(S2) The paper proposes an interesting new technique that, in some ways, seems to perform very well in experiments.


WEAKNESSES:
	(W1) The motivation for the technique in the sense of being for single-shot learning is not clear. Learning a distance metric between behaviors would seem to be applicable for learning from _many_ demonstrations -- in what way is the proposed approach specialized for single-shot learning?
	(W2) The presentation of the experimental results is, at times, incomplete and confusing. For example:
		(a) The ablation study shows lines for random/max sequence lengths, but (as best I can tell), the methods that generated these curves are never discussed.
		(b) The ablation study presented in Figures 5a and 5b generates more questions that it answers. For example, how is it that "(EESP) + pretrain" gets the lowest loss in 5b but does very poorly in 5a?
		(c) The experimental setup is unclear to the point where its doubtful that a reader could reproduce the results. In particular, at the beginning of Section 5, it's not clear exactly how the demonstrations are generated. The authors state that the simulation environments are "new and have all been updated to take motion capture data and produce view video data that can be used for training RL agents," but do not describe in any detail how the mocap data was processed to actually generate the demonstration video data.


RECOMMENDATION STATEMENT:
While the paper considers and important problem and may indeed describe a very promising new approach, the current experiments section is in need of a major rework before the story is clear.


QUESTIONS FOR AUTHORS:
	(Q1) How, exactly, were the demonstrations from mocap data produced?


MINOR COMMENTS:
	(MC1) The authors need to state in the abstract that their experiments are done on _simulated_ agents.
	(MC2) The authors state in the introduction that "the fundamental problem of imitation learning is how to align a demonstration in space and time with the agent's own space." Respectfully, don't think this is true. It is perhaps "a" fundamental problem, but certainly not "the" fundamental problem. Even with aligned state spaces, the general problem imitation learning poses several other challenges such as how to learn robust policies, how to incorporate multiple demonstrations if they are available, etc. The authors should revise the wording of this statement.
	(MC3) The caption of Figure 4 does not address (d)-(g).
	(MC4) The legend in Figure 4a says "GAILfO" but the text says the comparison algorithm is "GAIfO."
	(MC5) The text in Figures 4 and 5 is so small that it is unreadable. The figures need to be reformatted to address this.
	(MC6) The authors should reference and discuss how their work relates to / differs from a very similar recent paper [https://arxiv.org/pdf/1909.13392.pdf]

---

> ### Author Response · Authors · 2020-11-19
> **Good feedback, added more experiments & clarification on mocap data**
>
> Dear Reviewer 1,
>
> We appreciate your effort and suggestions, and thank you for recognizing this as “very promising new approach”.
>
> One of your concerns was regarding a desire for us to provide more of the details around our use of the mocap data used to generate renderings of a humanoid avatar. We are using the mocap data from the “CMU Graphics Lab Motion Capture Database” from 2002 (http://mocap.cs.cmu.edu/). To be thorough we provide the process at length. This data has been preprocessed to map the mocap markers to a human skeleton. Each recording contains the positions and orientations of the different joints of a human skeleton and can therefore directly be used to animate a humanoid mesh. This is a standard approach that has been widely used in prior literature like [Gleicher 1998, Rosales 2000, Lee 2002]. To be precise: at each mocap frame, the joints of a humanoid mesh model are set to the positions and orientations of their respective values in the recording. If a full humanoid mesh is not available, it is possible to add capsule mesh primitives between each recorded joint. This 3D mesh model is then rendered to an image through a 3rd person camera that follows the center of mass of the mesh at a fixed distance. We have added this description to the paper. Let us know if there are any more questions you have on this process.
> We hope that this sufficiently addresses the reviewer’s main question/concern.
>
> Allow us to address your other concerns:
>
> - W1: What we mean by “1-shot imitation” is that our method is able to learn to imitate an expert based on a single demonstration. Multiple previous works that _also_ address imitation learning from a single expert demonstration have also called their method “one-shot imitation” (see e.g. [Finn 2017] or [Yu 2018]).
> But since other reviewers also had concerns about this terminology, we have updated the paper and removed all instances of “one-shot imitation” and added a description of the problem.
> - W2:
> (a/b) In the editing process, the explanation of the ablations (random and max) got cut accidentally. By random and max we are referring to changing the distribution of sequence lengths used during training. We found that by focusing more on shorter length sequences learning improved over using a random sequence length or using the longest sequence possible (length = max episode time)  We’ve added this content to the paper. “(EESP) + pretrain” getting low error but poor reward, We found that the pretraining process often caused the model to overfit the data. As a result, this does not provide a good reward function for training. Training online allows the method to collect extensive data to train the model and prevents overfitting.
> (c) See comment on mocap data above.
> - MC3-5: We will update Fig.4-5 as well as their captions, thanks for bringing this to our attention. We have currently added new experiments to Fig. 4 but we will give them a proper full revision in terms of font size and style before the final draft.
> - MC6: This method uses the VGGNET model to output an encoding and uses that for computing distances. This paper is investigating a different problem setting where it is possible to query humans to compare the video demonstrations with agent demonstration. We have added this content and reference to the paper.
>
> Please let us know if this sufficiently addresses your concerns that prevented you from recommending this paper for acceptance.
>
> For your convenience, here's the link to the side-by-side comparison with the previous iteration of the paper: https://openreview.net/revisions/compare?id=MBdafA3G9k&left=Nqvt8Vplgz&right=cHuGdDlXGd
>
> #### References:
>
> - **[Gleicher 1998]** Gleicher, Michael. "Retargetting motion to new characters." Proceedings of the 25th annual conference on Computer graphics and interactive techniques. 1998.
> - **[Lee 2002]** Lee, Jehee, et al. "Interactive control of avatars animated with human motion data." Proceedings of the 29th annual conference on Computer graphics and interactive techniques. 2002.
> - **[Rosales 2000]**, Rosales, Rómer, and Stan Sclaroff. "Learning and synthesizing human body motion and posture." Proceedings Fourth IEEE International Conference on Automatic Face and Gesture Recognition (Cat. No. PR00580). IEEE, 2000.
> - **[Finn 2017]** Finn, Chelsea, et al. "One-shot visual imitation learning via meta-learning." arXiv preprint arXiv:1709.04905 (2017).
> - **[Yu 2018]** Yu, Tianhe, et al. "One-shot imitation from observing humans via domain-adaptive meta-learning." arXiv preprint arXiv:1802.01557 (2018).

---

> > ### Comment · AnonReviewer1 · 2020-11-24
> > **Thanks**
> >
> > Thanks to the authors for the response and the revisions to the paper. The paper has been improved  by the inclusion of the missing discussion relating to the ablation studies and the extra detail regarding the preprocessing of the mocap data.
> >
> > Unfortunately, I'm still a little bothered by the experiments section; especially Figure 4. The authors have identified several experimental domains ("humanoid 2d walk," "dog 2d," etc.) and several algorithms to evaluate (VIRL, GAILfO, GAIfO, TCN, etc.), but have chosen only to compare VIRL to a _subset_ of these methods in each subfigure. In particular, GAIfO was dropped entirely from (e)-(f). Why?

---

> > > ### Author Response · Authors · 2020-11-25
> > > **Response**
> > >
> > > Thank you for your feedback.
> > >
> > > We did not run GAIfO on (e-f) because we prioritized comparing VIRL with TCN that has shown to be much more capable in the problem setting of imitation from a single demonstration. Given the data so far it is unlikely that GAIfO will perform better on the much more challenging humanoid3D tasks in (e-f). However, we would be happy to run GAIfO on these tasks and include that data in the paper.

---

> > > > ### Comment · AnonReviewer1 · 2020-11-25
> > > > **Please Do**
> > > >
> > > > Yes, I think that would be appropriate. As it stands, the figures seem incomplete.

---

> ### Author Response · Authors · 2020-11-23
> **Feedback**
>
> Hello, please let us know if you have any further questions or comments on the paper.

---

### Official Review · AnonReviewer4 · 2020-10-28
**Good results in a challenging setting: one-shot imitation from visual demonstration**

**Rating:** 6
**Confidence:** 4

**Review:**

The approach described in the paper uses a relatively complicated architecture, with multiple losses and a very processed training set, but it achieves strong results when compared to two of the most similar published methods - TCNs (Time-Contrastive Networks) and GAIfO (Generative adversarial imitation from observations). A set of ablations is presented which is not exhaustive, but which is adequate to understand the comparative importance of the different parts of the approach. Overall, I think that there are some rough edges to the paper which could be improved, but the contribution of the paper is enough to warrant publication.

One problem is that the method is poorly explained in some places. Some of the terms need to be explained earlier in the paper - Siamese Network or Siamese Loss is never properly explained and TCN is used in the abstract without reference or definition. At the beginning of section 3, it is stated that the 'Siamese network triplet loss' is used in the proposed method, but Eq. 3 shows a contrastive loss with margin, not a triplet loss. Adding to the problems with clarity, there are a number of obvious typos which are distracting to the reader and make it clear that this is not a fully polished submission:

typos - oultine -> outline; Advisarial -> Adversarial; primariliy -> primarily; subsquent -> subsequent; independanlty -> independently; kinematiccally -> kinematically
fragments: 'Including recent Sim2Real quadreped robots and a huanoid with 38 DoF, which is a
particularly challenging problem domain.'; 'Where states and actions are discrete.'

The paper claims that the method can train an agent to do imitations from noisy visual data from single demonstrations, but this is not clearly shown by the experiments. The domains where the agent performs well (judging by the videos available on the linked website) are simple and cyclic and have little or no variation. It is unclear whether the agent would be able to 1-shot imitate a novel demonstration which was not within the training set already. On the more challenging domains, the videos show that the agent imitates the expert for a very short timespan before diverging. However, visual imitation is very difficult and the proposed method does perform better than the other baselines.

The paper would have more impact if the authors had shown whether the trained approach could be used for better performance or learning speed on new tasks where expert data was not available, perhaps by training a new policy on top of the LSTM, or distilling to a new agent and then finetuning.

Overall, the paper is exciting because it shows the value of the complex architecture for solving a challenging problem. The experiments are convincing and show that this is a promising approach which may be useful in a real-world domain.

Pros:
- The method is well-designed and is a natural extension on existing work (TCNs, Siamese nets, GAIfO). There is enough novelty for the work to be published.
- The results show that the approach works well on a number of different continuous control environments
- The comparisons to other published methods, the baselines and ablations, are well chosen.
- The supplementary contains additional details on training, and additional analysis, which is valuable

Cons:
- The text is poorly written in places - Acronyms and terms need to be explained, and spelling and grammar need to be proofed. However, this is an easy fix that should not prevent the paper from being published.
- The authors have not adequately explained whether the model is actually capable of 1-shot imitation. It is not clear from the domains whether the agent has simply memorized the full data distribution.
- The model is limited if there is no way to reuse the model without demonstrations.

---

> ### Author Response · Authors · 2020-11-19
> **Feedback greatly appreciated. Paper updated accordingly.**
>
> Dear Reviewer 4,
>
> Thanks for your time and effort.
> We found your feedback valuable and have taken the following into account:
>
> - “Acronyms and terms need to be explained” - We’ve updated the paper to explain all acronyms and cite them properly. We’ve also made another pass on orthography because quite a few errors slipped through, as you rightfully pointed out.
> - “The authors have not adequately explained whether the model is actually capable of 1-shot imitation” - We’ve updated the text to reflect the fact that we mean we’re able to learn imitation from a single expert demonstration. Some reviewers found “one-shot imitation” a problematic term for this and so we’ve removed it from the text and made our intention clearer.
> - We apologize for orthographic and acronym mistakes. We’ve made a full pass on the paper, incorporating spelling fixes you suggested and making sure all acronyms are explained and cited.
>
> Please have a look and see if these changes address your concerns.
>
> Re “The model is limited if there is no way to reuse the model without demonstrations”:
> We usually use our method to fit a single behavior without goal-conditioning. That means yes, for each new behavior, we have to retrain our model. But this is the same for TCN and GAIfO. We do use multi-task training in some examples for the humanoid3d environment with the intent that training across multiple tasks improves usability across tasks.
>
> For your convenience, here's the link to the side-by-side comparison with the previous iteration of the paper: https://openreview.net/revisions/compare?id=MBdafA3G9k&left=Nqvt8Vplgz&right=cHuGdDlXGd

---

### Author Response · Authors · 2020-11-18
**Problem setting clarification and difference to prior works**

There appears to be some confusion over the problem setting in the paper. We hope to fix this by first providing a better outline of the most related work with respect to the assumptions those models make and then describe our method and how the data is used in more detail. There has been significant work in the area of imitation learning, here we organize the related work into groups depending on the assumptions of the methods used for producing imitative behaviour in order to emphasize the importance of our method and the desired application to a more realistic scenario under assumptions similar to the real world. Starting with methods that require access to an expert policy, training in the same simulation, to provide both state and action information (GIAL, infoGAIL, etc). Some recent methods relax the need for access to the expert actions (Behaviour cloning from observation, GAILfO, etc). The next level of imitation method can use images instead of states, taking us a step closer to being used in the real world using camera information (T-Rex, D-Rex, etc). However, these methods still require lots of data from a policy trained on the agent in the same simulation with the same dynamics, including ranking information. We relax all of these assumptions, similar to the TCN work but while requiring far less data (as low as a single example motion), and learn imitative behaviour from noisy image data from another agent. We will rewrite the related work section to emphasize this relationship clearly.

There have been many questions as to how the training works and how the data is used, we hope to clear up this confusion and outline just how difficult the learning problem is that we address in the paper. For a single task, we take a clip of motion from the CMU motion capture database. We use this data to animate a character in the simulation in order to record video data from a similar point of view with similar lighting as the agent in the simulation by kinematically setting the joint angels of the simulated “expert” (we do not assume this data is from an expert policy, it just needs to be a recorded motion). At the beginning of an episode, we can start the agent and the imitation motion and animate the “expert” agent forward in time, capturing images from both, so that this video data can be used to train the reward function and compute rewards as the distance between the images sequences created. The rendered kinematic agent can be seen in the background of most videos on the provided website (https://sites.google.com/view/virl1). We also provide state inputs to the policy, not image inputs, because in the real world a robot will not have access to the privileged 3d person visual information for itself. This is a very rare imitation setting that has not seen much attention due to its difficulty and most algorithms (GAIL, D-Rex, etc) are not designed to work in this setting.

Individual responses to follow.

---

### Decision · Program_Chairs · 2021-01-07
**Final Decision**

**Decision:**

Reject

**Comment:**

The paper considers the problem of learning to imitate behaviors from visual demonstrations, without access to expert actions. Consistent with recent approaches, the proposed method uses a neural network to measure the similarity between visual demonstrations and the agent's behavior, and employs this metric as a reward in RL. The primary contribution is the use of a recurrent siamese network that is trained to measure the distance between motions, as a means of better dealing with the challenges of imitation learning from a small number of (as few as one) noisy visual demonstrations. Experiments on a variety of simulated domains show that the proposed approach achieves reasonable results.

The paper was reviewed by four knowledgeable referees, who read the author responses and engaged in extensive discussion. The reviewers agree that learning to imitate behaviors from a small amount of noisy demonstrations is a challenging and important problem that is of significant interest. The proposed method nicely extends existing approaches to visual imitation learning, and the results reveal that the method performs well in a variety of continuous control domains. The reviewers raise several concerns regarding the clarity of the technical presentation and the sufficiency of the experimental evaluation. The authors have made a significant effort to address these concerns in their responses and updates to the paper, which the reviewers very much appreciate. However, some of the reviewers' primary concerns regarding clarity and the thoroughness of the experimental evaluation remain. This work has the potential to make a really nice contribution and the authors are encouraged to take this feedback into account for any future version of the manuscript.